# Isolation of an Acidophilic Cellulolytic Bacterial Strain and Its Cellulase Production Characteristics

**Shijia Zhang [1], Zhichao Wang [1], Jiong Shen [1], Xuantong Chen [2] and Juan Zhang [1,\*]**

[1]   School of Civil Engineering Architecture and Environment, Hubei University of Technology,
Wuhan 430068, China; 102010749@hbut.edu.cn (S.Z.); 2110651308@hbut.edu.cn (Z.W.);
102200854@hbut.edu.cn (J.S.)

[2]   Department of Biology, Lakehead University, Thunder Bay, ON P7B 5E1, Canada; xchen24@lakeheadu.ca

\*   Correspondence: zhj198100@163.com

**Abstract:** The aim of the study was to isolate and identify a highly efficient cellulolytic bacteria strain that can be used in acidic environments, and then investigate its cellulase production characteristics for the effective utilization of agricultural waste. For this purpose, we set a series of isolation and screening steps, 21 strains were isolated from soil, and an acidophilic strain labeled as B13-2 with high cellulase production was screened using the Gram' iodine method and cellulase activity assay; it was identified as *Raoultella terrigena*. Lastly, the culture conditions such as incubation time, incubation temperature, pH, carbon sources, nitrogen sources, and inoculum size were optimized via single-factor experiments, and on this basis, the cellulase production of strain B13-2 was optimized using response surface methodology with cellulase activity as the optimization goal. The results of the response surface optimization showed that the optimum incubation time is 3.1 days, the optimum temperature is 29.9 °C, the optimum pH is 4.1, and the optimum inoculum size is 1.50%, the cellulase activity reached a maximum of 13.503 U/mL, which was about 140% higher than that before optimization. In particular, strain B13-2 had higher cellulase production when rice straws were used as the natural carbon source. Meanwhile, the SEM pictures demonstrated that the surface of the substrate rice straws in an acidic buffer with strain B13-2 was uneven, with larger holes than in the neutral buffer after incubation. It further proved that this strain has a stronger ability to degrade cellulose under acidic conditions. The B13-2 is a kind of acidophilic cellulolytic bacteria. Therefore, it has the potential to be developed into a silage additive agent and provides a high-quality strain resource for the high-value biotransformation of agricultural waste and lays a certain foundation for the sustainable development of agricultural cultivation.

**Keywords:** acidophilic cellulolytic bacteria; rice straws; cellulase; optimization

## 1. Introduction

Currently, the most abundant and cheapest renewable resource in the world is cellulose, however, the vast majority of cellulose has not yet been fully utilized [1]. For example, as for agricultural waste, the main source of cellulosic materials, the total output of crop straws, is up to 700 million tons per year in China [2]. In recent years, scientists have been exploring multiple approaches for the development and resource recycling of cellulosic materials. In order to utilize agricultural waste in an eco-friendly way, it is necessary to convert the cellulosic polysaccharides into fermentable sugars for utilization [3]. In this process, the difficulty is that natural cellulosic materials contain a large amount of cellulose (crystalline and amorphous) with hemicellulose and lignin which is difficult to be decomposed and utilized [3]. Therefore, the materials need to be pretreated. We usually use physical, chemical, and biological treatment to degrade cellulose. Among these treatments, biological treatment (cellulolytic microorganisms and enzymes) is the most environmentally friendly means to solve this challenge [4].



Cellulases are a complex enzyme family composed of a combination of hydrolases, which can effectively degrade natural cellulose into oligosaccharides [5]. The main sources of cellulases in nature are fungi, bacteria, actinomycetes, and some protozoa [6]. Among them, studies targeting bacteria are becoming a new hot topic. Compared with fungi cells, the bacteria cells can secrete more complex cellulases, with abundant ingredients, which are more likely to exert synergistic effects between enzyme families and are more beneficial for cellulose degradation. Bacteria are more adaptable to the environment and the enzymes that are secreted are more stable [7]. There are many reports on the screening of cellulolytic bacterial strains, and most of the optimal pH conditions for cellulase production are neutral. Carlos et al. [8] isolated a strain of *Stenotrophomonas maltophilia* from the soil, and after single-factor optimization, the maximum cellulase activity was 0.082 U/mL while the initial pH was 6.3, the carbon source concentration was 0.72% and the nitrogen source concentration was 1.5%. Sun et al. [9] isolated a strain of *Bacillus licheniformis* from manure with the highest enzyme activity (21.96 U) at an initial pH of 6.5. Du et al. [10] optimized the cellulase activity of strain *Alcaligenes faecalis* DL-08 by response surface methodology, and the results showed that the cellulase activity of the strain could be increased by 5.6% when the strain was cultured in a neutral medium. At present, most cellulolytic bacteria are reported to be more productive under neutral conditions, and few strains have high enzyme production capacity under acidic conditions.

Acidophilic cellulolytic bacteria play an important role in biomass processing. It has been found that the addition of acidophilic cellulase preparation or inoculant to silage will not affect the fermentation process of silage, but can also have the effect of breaking the cellulose cell walls, and increasing the content of soluble carbohydrates as well as digestibility [11]. Meanwhile, it can consume oxygen, increase the degree of anaerobiosis, promote the growth and reproduction of lactic acid bacteria, and avoid abnormal fermentation. Cellulase or bacterial preparations are often used in combination with lactic acid bacteria additives to improve the fermentation quality of silage [12,13]. Huang et al. [12] isolated three acidophilic cellulolytic bacterial strains with high cellulase activity and applied them to *Pennisetum* silage, and the results showed that the addition of cellulolytic bacterial suspensions increased the lactic acid content of the silage and contributed to the success of silage. It also reduced the content of NDF (neutral detergent fiber), ADF (acid detergent fiber), and HC (hemicellulose), which improved the silage quality to different degrees. Meng et al. [14] isolated six acidophilic cellulose-degrading bacterial strains and the strain Gs9 identified as *Lactobacillus buchneri* has the potential to be used in the development of silage microbial agents. In the present study, an acidophilic strain of cellulolytic bacteria was isolated from soil from the riverside, and further culture conditions were optimized to achieve maximum cellulase production, and the study will provide a reference for further production of high-quality silage and effective utilization of agricultural waste.

## 2. Materials and Methods

### 2.1. Sample Source

The sample source was humus soil along the Xunsi River in Wuhan, Hubei province. The geographical locations of the sampling site are shown in Table 1.

**Table 1.** Geographical locations of the sampling site.

| Sample Number | Latitude | Longitude |
| --- | --- | --- |
| A11-2 | 30°29′39″ | 114°18′24″ |
| A13-2 | 30°29′39″ | 114°18′24″ |
| B12-4 | 30°29′6″ | 114°18′45″ |
| B13-2 | 30°29′6″ | 114°18′45″ |
| M2-12 | 30°28′48″ | 114°18′45″ |

### 2.2. Culture Media

According to the related literatures [15,16], we prepared different culture media for bacterial growth and cellulase production. These culture media include (A) Reasoner's 2A (R2A) agar, (B) Luria–Bertani (LB) broth, (C) carboxymethyl cellulose (CMC) agar, and (D) cellulase production broth. Their compositions were as follows:

(A)  R2A agar: 0.5 g yeast extract, 0.5 g peptone, 0.5 g starch, 0.5 g $MgSO_4$, 0.5 g casein hydrolysate, 0.5 g glucose, 0.3 g $K_2HPO_4$, 0.3 g sodium pyruvate, 15 g agar, and distilled water up to 1 L.
(B)  LB broth: 10 g peptone, 5 g yeast extract, 5 g NaCl, and distilled water up to 1 L.
(C)  CMC agar: 5 g CMC, 1 g $NaNO_3$, 1 g $K_2HPO_4$, 1 g NaCl, 0.5 g $MgSO_4$, 0.5 g yeast extract, 15 g agar, and distilled water up to 1 L.
(D)  Cellulase production broth: 10 g CMC, 10.0 g peptone, 7.5 mg $FeSO_4 \cdot 7H_2O$, 2.5 mg $MnSO_4 \cdot H_2O$, 2.0 mg $ZnSO_4$, and distilled water up to 1 L.

### 2.3. Screening and Isolation

According to the relevant literatures [16,17], we designed the isolation, screening, and optimization process (Figure 1). Soil samples (5 g) were diluted with 100 mL of sterile water and then transferred into a shaking incubator for 60 min at 30 °C; After incubation, microbial suspension (50 μL) was plated onto R2A agar. The plates were incubated at 30 °C for 2 days. The incubation time can be shortened or extended according to the actual situation. According to their morphology, size and color, bacterial clones were selected for purification. These colonies were screened preliminarily for their ability to produce cellulase using the iodine method. For this purpose, the isolated colonies were grown in LB broth (5 mL) for 24 h shaking at 30 °C. All bacterial suspensions (10 μL) were singularly dropped onto the center of CMC agar plates and then incubated at 30 °C for 2 days. After incubation, the plates were stained with Gram's iodine as an indicator to visualize the cellulase activity [18]. The formation of a clear halo around the colonies confirms cellulase activity.

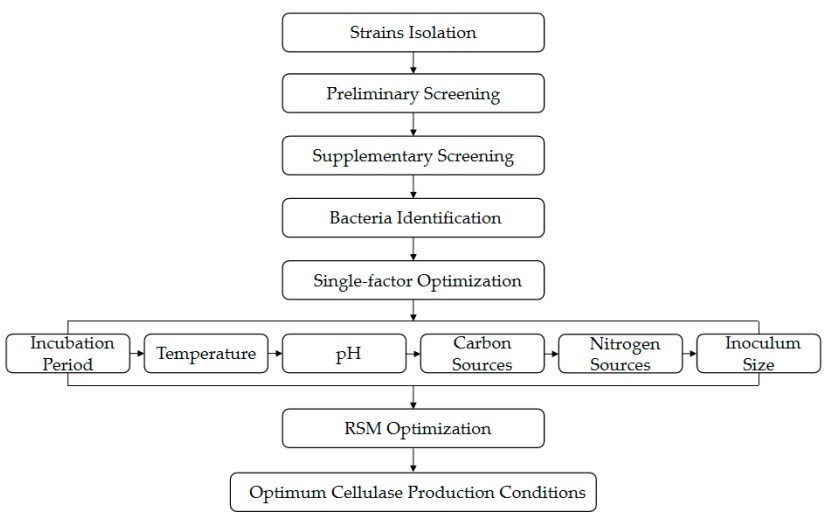

**Figure 1.** Isolation, screening, and optimization process of strain B13-2.

In order to reliably quantify the cellulase production capacity of different strains in acidic condition, it's necessary to make subsequent supplementary screening tests. The positive isolates were cultured in 5 mL of LB broth (24 h, 30 °C, and 150 rpm). A total of 1 mL of the bacterial suspension was centrifuged at $8000 \times g$ for 10 min and the cells were re-suspended in sterile water. Separately, the re-suspended cells were inoculated into a 250-mL conical flask with cellulase production broth (100 mL) containing citrate buffer (0.1 M, pH 5). Subsequently, the flasks were incubated at 30 °C and 150 rpm for 5 days After incubation, bacterial suspension was centrifuged at $8000 \times g$ and 4 °C for 10 min and

the supernatant was crude enzyme. Quantification of cellulase activity of the crude enzyme and the strain with higher activity was selected as the subsequent study object.

### 2.4. Bacteria Identification by 16S rDNA Sequencing

The 16S rDNA sequencing was performed to identify the species and genera of bacteria. The extraction of DNA from the strain was performed using the Ezup Column Bacteria Genomic DNA Purification kit (Sangon Biotech, Shanghai, China), according to the instructions given by the manufacturer. PCR amplification was performed using the bacterial universal primers 27F (5′-AGAGTTTGATCATGGCTCAG-3′) and 1492R (5′-TAGGGTTACCTT GTTACGACTT-3′). The PCR reaction system consisted of bacterial genomic DNA 0.5 μL, 10% of $Mg^{2+}$ × Buffer 2.5 μL, dNTP 1 μL, DNA polymerase 0.2 μL, and 10 μmol/L forward and reverse primers 0.5 μL, respectively. Then, add double distilled water to 25 μL. The PCR conditions were pre-denaturation at 94 °C for 4 min, 94 °C denaturation at 94 °C for 45 s, renaturation at 55 °C for 45 s, and extension at 72 °C for 1 min for a total of 30 cycles. The repair was extended for 10 min at 72 °C, and the reaction was terminated at 4 °C. The DNA quality and quantity were measured by agarose gel electrophoresis (using 1% agarose gel, electrophoresis was performed at a voltage of 150V and stopped after 20 min). The PCR amplification products were purified by the SanPrep Column DNA Gel Extraction kit and then sent to Sangon Biotech (Shanghai, China) Co., Ltd. For sequencing. The 16S rDNA gene sequencing results were compared to find similar sequences by BLAST comparison at NCBI. Some similar 16S rDNA sequences were selected from the GenBank nucleic acid database, and multiple sequences were compared using the Clustal X 1.83 software with the bootstrap value set to 1000. The phylogenetic tree was constructed using the neighbor-joining method in the software MEGA6.

### 2.5. Quantification of Cellulase Activity

Measuring the release of reducing sugars determined quantitative cellulase activities through the 3,5-dinitrosalicylic acid (DNS) method [19,20]. The crude enzyme (1 mL), 0.1 M citrate buffer (pH 4.8, 1 mL), and a filter paper strip (1 cm × 6 cm) were transferred into a test tube, and heated at 50 °C for 30 min. DNS solution (1.5 mL) was added to the mixture and the tube was heated at 100 °C for 5 min.

Using glucose as a standard for the calibration curve estimated the release of reducing sugars in the reaction mixture. The reaction mixture was measured at 540 nm by using an ultraviolet-visible spectrophotometer (SHIMADZU, Kyoto, Japan). The cellulase activity was determined in U/mL. One unit (U) was defined as the amount of cellulase that released 1 μg of reducing sugar per mL.

### 2.6. Quantification of Weight Loss Rate

The fermentation broth was filtered through filter paper and then rinsed repeatedly with distilled water until the filtrate became colorless. The cleaned filter residue was dried to a constant weight at 80 °C then weighed, and the weight loss rate was calculated. Three replicates were set up for each experiment.

### 2.7. Single-Factor Optimization of Cellulase Production Conditions

The effect of various factors such as incubation time, temperature, pH, and other culture conditions based on cellulase production broth was investigated for optimal cellulase production. The optimization process is shown in Figure 1. During the optimization process, the previous optimal factor obtained was used for the subsequent optimization of the next factor. The growth of the organism was assessed by measuring the optical density of bacterial suspension at 600 nm. Three replicates were set up for each experiment. Measuring the cellulase activity of crude enzyme or weight loss rate of carbon source determined cellulase production capacity.

### 2.7.1. Incubation Period

The cellulase production broth (100 mL) was incubated with cultured bacterial solution (1 mL) in a shaking incubator (150 rpm) for 5 days. Determining the cellulase activity at 24, 48, 72, 96, 120, and 144 h examined the effect of incubation time on cellulase production.

### 2.7.2. Temperature

The cellulase production broth (100 mL) was incubated with bacterial solution (1 mL) in a shaking incubator (150 rpm) at 20, 25, 30, 35, and 40 °C for optimal incubation time. Determining the cellulase activity examined the effect of temperature on cellulase production.

### 2.7.3. pH

The media with different pH buffers were prepared to ensure pH stability during the incubation process. The cellulase production broth (100 mL) was incubated with cultured bacterial solution (1 mL) in a shaking incubator (150 rpm) at the pH ranges of 3.5–8 at the optimal temperature for optimal incubation time. Determining the cellulase activity examined the effect of pH on cellulase production.

### 2.7.4. Carbon Sources

CMC was replaced with other carbon sources such as lotus seed, maize straws, wheat straws, wheat bran, rice straws, and rice husks. The cellulase production broth (100 mL) was incubated with a carbon source (1.0% $w/v$) and cultured bacterial solution (1 mL) in a shaking incubator (150 rpm) at the optimal temperature and pH for optimal incubation time. Determining the cellulase activity and weight loss rate determination examined the effect of carbon sources on cellulase production.

### 2.7.5. Nitrogen Sources

Various nitrogen sources such as yeast extract, peptone, sodium nitrate, ammonium sulfate, and urea were selected in cellulase production. The cellulase production broth (100 mL) was incubated with a nitrogen source (1.0% $w/v$) and cultured bacterial solution (1 mL) in a shaking incubator (150 rpm) at the optimal temperature and pH for optimal incubation time. Determining the cellulase activity and weight loss rate determination examined the effect of nitrogen sources on cellulase production.

### 2.7.6. Inoculum Size

The cellulase production broth (100 mL) was incubated with cultured bacterial solution (1~5 mL) in a shaking incubator (150 rpm) at the optimal temperature and pH for optimal incubation time. Determining the cellulase activity and weight loss rate determination examined the effect of inoculum size on cellulase production.

### 2.8. Optimization of Cellulase Production through Response Surface Methodology (RSM)

The optimal ranges of parameters for cellulase production were determined by preliminary single-factor experiments, and based on them a three-level, four-variable BBD was applied to statistically optimize the cellulase production conditions of the B13-2. The incubation time (A), temperature (B), pH (C), and inoculum size (D) were considered independent variables, and the cellulase activity was considered the response variable. The ranges and levels of the variables are shown in Table 2. In order to minimize the effects of unexpected variability in the observed responses, experimental runs were carried out randomly.

For the purpose of analysis of the experimental design and prediction of c optimized conditions, experimental data were analyzed via Design Expert 10.0.1 and fitted to a quadratic regression model. The model evaluated the effect of each independent variable to the response variable. Subsequently, three replicate experiments were conducted using the theoretically optimal incubation conditions to verify the accuracy of the model.

**Table 2.** Box–Behnken experiment design factors and levels.

| | | | Levels | |
|---|---|---|---|---|
| **Factors** | **Symbols** | **−1** | **0** | **1** |
| Time/d | A | 2 | 3 | 4 |
| Temperature/°C | B | 25 | 30 | 35 |
| pH | C | 3.5 | 4.0 | 4.5 |
| Inoculum size/% | D | 0.5 | 1.5 | 2.5 |

*2.9. Morphological Observation of Substrate Rice Straws before and after Incubation under Different pH Conditions*

Strain B13-2 was cultured for three days at pH 4.1 and pH 7.0. Rice straws were the only carbon source of the cellulase production broth.

Straw samples were collected from the broth and fixed overnight at 4 °C in 25% glutaraldehyde. After fixation, the samples were gradually dehydrated with ethanol at 30%, 50%, 70%, 80%, 90%, and 100% concentrations and freeze-dried. The samples were then mounted on an aluminum sample holder covered with a layer of gold and observed with an ultra-high resolution field emission scanning electron microscope (HITACHI S4800, Tokyo, Japan), and imaged at a magnification of 500 times.

## 3. Results

*3.1. Screening and Identification*

Twenty-one strains of bacteria were isolated from the collected soil samples and cultured on CMC agar. After incubation, the plates were stained with Gram's iodine, and five strains with clear halos (Figure 2) were selected. The presence of halos indicated that these five strains have cellulose-degrading capacity.

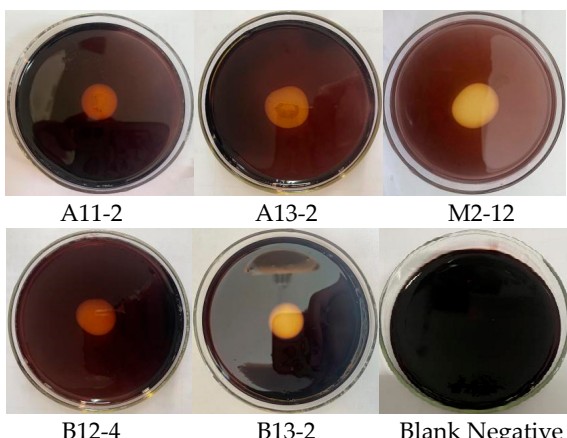

**Figure 2.** Gram's iodine staining results.

Supplementary screening tests were carried out for those five cellulose-degrading bacteria obtained from the preliminary screening. After incubation in acidic condition, the crude cellulase solution was collected from cellulase production broth and the cellulase activity of each strain was measured (Table 3). The cellulase activity of the strain labeled as B13-2 was the highest at 4.83 ± 0.06 U/mL in acidic condition. Subsequently, the cellulase production conditions of strain B13-2 were optimized.

**Table 3.** Cellulase activity of cellulolytic bacteria strains in supplementary screening tests.

| Strain | Cellulase Activity (U/mL) |
|--------|---------------------------|
| A11-2 | $4.01 \pm 0.16$ |
| A13-2 | $4.28 \pm 0.09$ |
| B12-4 | $4.14 \pm 0.13$ |
| B13-2 | $4.83 \pm 0.06$ |
| M2-12 | $3.73 \pm 0.10$ |

### 3.2. Molecular Identification of the Cellulase-Producing Strain

The agarose gel electrophoretogram of B13-2 16S rDNA was shown in Figure 3. The results showed that the full-length 16S rDNA of B13-2 (NCBI accession number: OR123418) was 1476 bp, and a phylogenetic tree was constructed using neighbor-joining (Figure 4). The phylogenetic analyses indicated that the identity of the B13-2 strain was *Raoultella terrigena* strain with a 98% similarity. Combining the colony morphology and molecular biology results of this strain, we identified the cellulose-degrading strain from soil samples belonging to genera *Raoultella terrigena*.

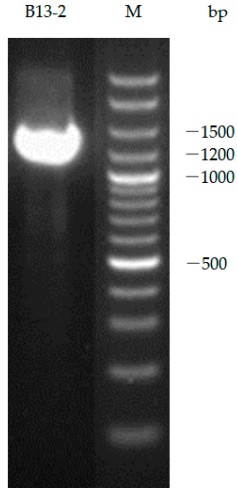

**Figure 3.** Agarose gel electrophoretogram of B13-2 16S rDNA.

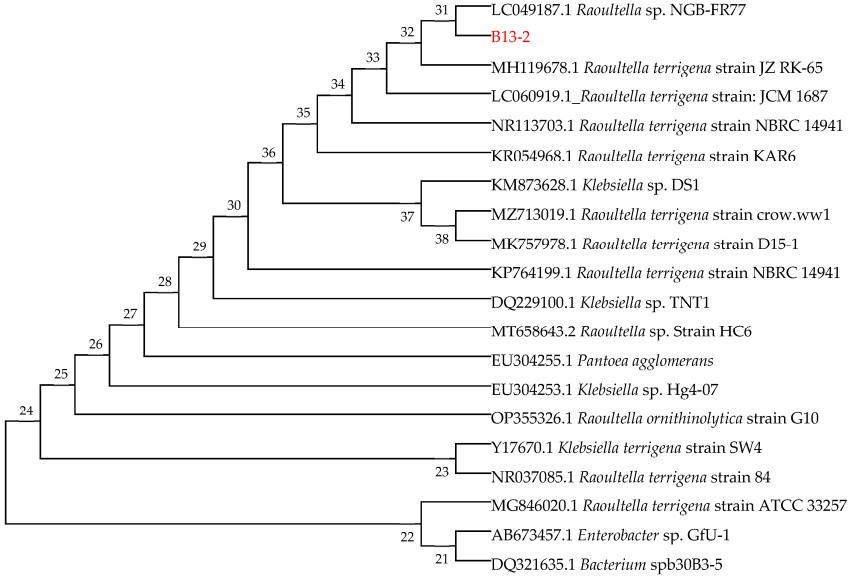

**Figure 4.** Phylogenetic tree constructed based on cellulose-degrading bacteria B13-2 16S rDNA.

### 3.3. Preliminary Single-Factor Experiments

3.3.1. Effect of Incubation Time on Cellulase Production

The cellulase production of strain B13-2 at different incubation times is shown in Figure 5. During the incubation, the cellulase activity increased continuously as the fermentation proceeded and peaked at $(5.90 \pm 0.15)$ U/mL then decreased rapidly after 3 days, indicating that the cellulase production stopped or decreased significantly.

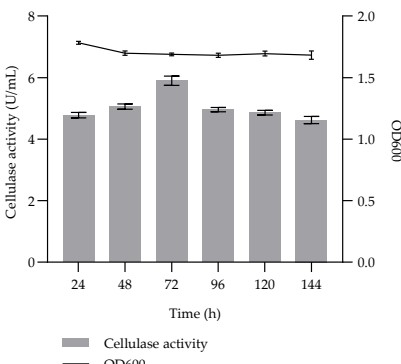

**Figure 5.** Effect of incubation time on the cellulase production capacity of strain B13-2.

3.3.2. Effect of Temperature on Cellulase Production

Temperature is a crucial parameter affecting microbial enzyme production [21]. The cellulase production of the strain B13-2 incubated at different temperatures showed a trend of increasing and then decreasing with the increase in temperature, reaching the highest of $(7.06 \pm 0.09)$ U/mL when the incubation temperature was 30 °C (Figure 6). In addition, strain B13-2 produced considerable cellulase when the temperature was in the range of 20~45 °C, indicating that the strain has a relatively wide range of growth temperatures and can adapt to different environmental temperatures.

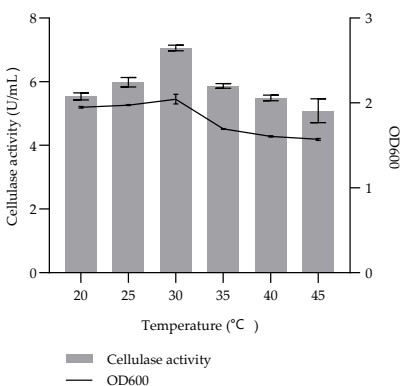

**Figure 6.** Effect of temperature on the cellulase production capacity of strain B13-2.

3.3.3. Effect of pH on Cellulase Production

During the whole process of enzyme production in liquid fermentation, the pH of bacterial suspension will be changed due to the metabolism of the bacteria. Therefore, this experiment uses a buffer system to control the pH of bacterial suspension so that the microorganisms can produce enzymes in a stable pH environment during the whole fermentation process. The effect of different pH on the cellulase production of strain B13-2 is shown in Figure 7. The control of pH during fermentation is necessary to improve the productivity of cellulase [17]. When the pH was 4.0, the strain B13-2 produced the highest cellulase production (cellulase activity = $11.36 \pm 0.15$ U/mL) after 3 days of incubation. Strain B13-2 was more inclined to secrete cellulase in an acidic environment, in addition, when the pH increased to 8.0, the strain still had good cellulase production capacity.

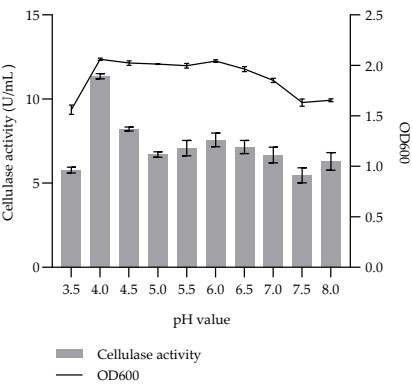

**Figure 7.** Effect of pH on the cellulase production capacity of strain B13-2.

### 3.3.4. Effect of Carbon Sources on Cellulase Production

The carbon sources selected in this experiment were CMC and six natural cellulose materials which were lotus seedpods, maize straws, wheat straws, wheat bran, rice straws, and rice husks. When rice straws were used as the natural carbon source, the cellulase production of strain B13-2 was the highest (cellulase activity = 11.89 ± 0.50 U/mL) after three days of incubation, and the weight loss rate of the carbon source was the highest at the same time, reaching 24.14% (Figure 8).

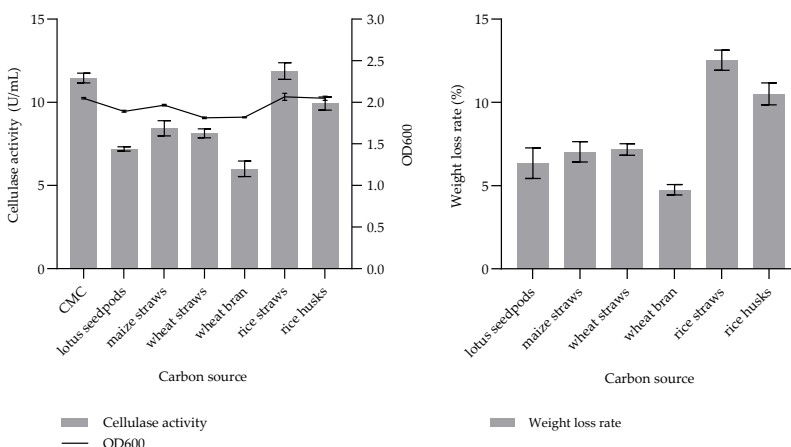

**Figure 8.** Effect of carbon source on the cellulase production capacity of strain B13-2.

### 3.3.5. Effect of Nitrogen Sources on Cellulase Production

In this experiment, five materials were used as nitrogen sources for cellulase production, which were yeast extract, peptone, sodium nitrate, ammonium sulfate, and urea. When peptone was used as the nitrogen source, the strain produced the highest cellulase production (cellulase activity = 11.76 ± 0.14 U/mL) after three days of incubation, and the weight loss rate of the carbon source was the highest at this time, at 24.14% (Figure 9).

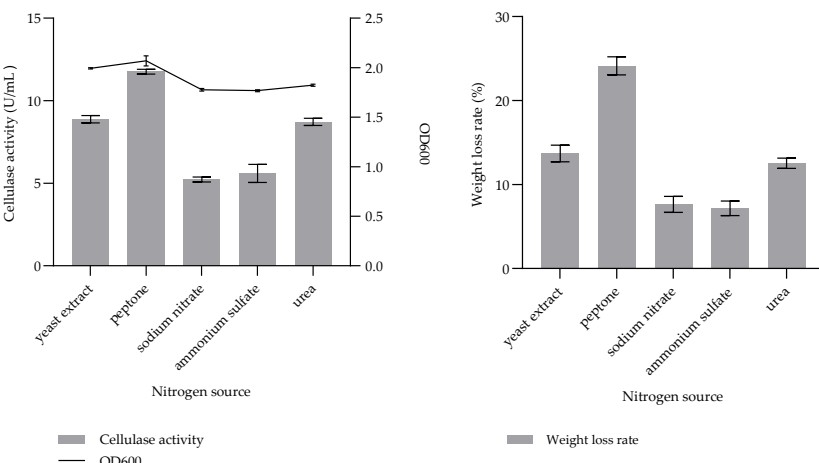

**Figure 9.** Effect of nitrogen source on the cellulase production capacity of strain B13-2.

### 3.3.6. Effect of Inoculum Size on Cellulase Production

As shown in Figure 10, 1% and 2% inoculation size were better for cellulase production of strain B13-2 with cellulase activity and weight loss of (11.87 ± 0.16) U/mL, 24.14% and (12.87 ± 0.11) U/mL, 25.17 %, respectively.

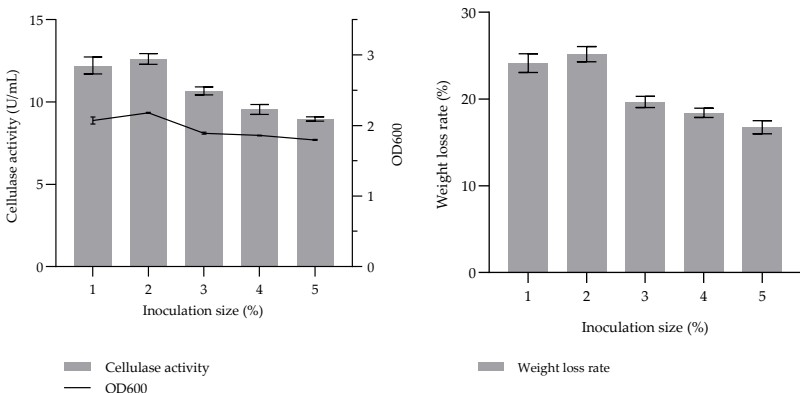

**Figure 10.** Effect of inoculum size on the cellulase production capacity of strain B13-2.

### 3.4. Optimization of the B13-2 Cellulase Production Using Response Surface Methodology (RSM)

According to the preliminary range of the parameters determined by the single-factor test for cellulase production, rice straws were taken as the carbon source, peptone as the nitrogen source, time (A), temperature (B), pH (C) and inoculum size (D) were selected as variables, and the cellulase activity was taken as the optimization target. The cellulase production conditions were fitted by multiple regression using the software Design Expert 10.0.1.

A total of 29 runs of the BBD and response based on the experimental runs are shown in Table 4.

**Table 4.** Treatment combination and response value of each factor level.

| Run | A Time/d | B Temperature/°C | C pH | D Inoculum Size/% | Cellulase Activity/(U/mL) |
|-----|----------|------------------|------|-------------------|---------------------------|
| 1 | 3 | 35 | 4 | 2.5 | 9.58 |
| 2 | 3 | 25 | 4 | 2.5 | 9.80 |
| 3 | 4 | 30 | 4 | 2.5 | 10.01 |
| 4 | 3 | 35 | 4 | 0.5 | 9.56 |
| 5 | 3 | 35 | 4.5 | 1.5 | 9.06 |
| 6 | 3 | 25 | 3.5 | 1.5 | 5.93 |
| 7 | 3 | 30 | 4 | 1.5 | 13.60 |

**Table 4.** *Cont.*

| Run | A Time/d | B Temperature/°C | C pH | D Inoculum Size/% | Cellulase Activity/(U/mL) |
|-----|----------|------------------|------|-------------------|---------------------------|
| 8 | 4 | 30 | 4.5 | 1.5 | 9.10 |
| 9 | 2 | 25 | 4 | 1.5 | 8.70 |
| 10 | 2 | 30 | 4.5 | 1.5 | 8.82 |
| 11 | 4 | 35 | 4 | 1.5 | 9.60 |
| 12 | 4 | 30 | 4 | 1.5 | 9.61 |
| 13 | 3 | 30 | 4.5 | 1.5 | 9.10 |
| 14 | 4 | 30 | 3.5 | 1.5 | 5.86 |
| 15 | 4 | 25 | 4 | 1.5 | 9.73 |
| 16 | 3 | 30 | 4 | 1.5 | 13.07 |
| 17 | 3 | 30 | 4.5 | 2.5 | 8.60 |
| 18 | 3 | 25 | 4 | 0.5 | 9.80 |
| 19 | 3 | 35 | 3.5 | 1.5 | 5.58 |
| 20 | 2 | 35 | 4 | 1.5 | 9.46 |
| 21 | 3 | 30 | 4 | 1.5 | 13.29 |
| 22 | 2 | 30 | 4 | 2.5 | 9.41 |
| 23 | 2 | 30 | 4 | 0.5 | 9.62 |
| 24 | 3 | 30 | 4 | 1.5 | 13.42 |
| 25 | 3 | 30 | 3.5 | 0.5 | 5.07 |
| 26 | 3 | 25 | 4.5 | 1.5 | 9.32 |
| 27 | 3 | 30 | 3.5 | 2.5 | 6.52 |
| 28 | 2 | 30 | 3.5 | 1.5 | 5.30 |
| 29 | 3 | 30 | 4 | 1.5 | 13.59 |

### 3.4.1. Analysis of Variance (ANOVA)

Table 5 shows the analysis of variance (ANOVA) for the fitted quadratic polynomial model of cellulase production. We can see from Table 5, a model F-value of 176.11 and a low probability value (*p*-value < 0.0001) show that the model was significant for cellulase production capacity. Values of P greater than 0.05 indicate that model terms are not significant, while values less than 0.05 indicate that the model terms are significant [22]. "Adeq Precision" measures the signal to noise ratio. A ratio greater than four is desirable.

**Table 5.** Analysis of variance in response surface mode.

| Source | Sum of Squares | df | Mean Square | F Value | *p*-Value |
|--------|----------------|----|----|---------|-----------|
| Model | 162.29 | 14 | 11.59 | 176.11 | <0.0001 |
| A-Time | 0.32 | 1 | 0.32 | 4.91 | 0.0437 |
| B-Temperature | 0.015 | 1 | 0.015 | 0.22 | 0.6442 |
| C-pH | 31.59 | 1 | 31.59 | 479.93 | <0.0001 |
| D-inoculum size | 8.1 | 1 | 8.1 | 123.07 | <0.0001 |
| AB | 0.2 | 1 | 0.2 | 3.03 | 0.1037 |
| AC | 0.021 | 1 | 0.021 | 0.32 | 0.5815 |
| AD | 0.095 | 1 | 0.095 | 1.44 | 0.2495 |
| BC | $2.36 \times 10^{-3}$ | 1 | $2.36 \times 10^{-3}$ | 0.036 | 0.8526 |
| BD | $2.03 \times 10^{-4}$ | 1 | $2.03 \times 10^{-4}$ | $3.09 \times 10^{-3}$ | 0.9565 |
| CD | 0.95 | 1 | 0.95 | 14.46 | 0.0019 |
| A2 | 25.36 | 1 | 25.36 | 385.21 | <0.0001 |
| B2 | 22.56 | 1 | 22.56 | 342.67 | <0.0001 |
| C2 | 109.03 | 1 | 109.03 | 1656.47 | <0.0001 |
| D2 | 20.91 | 1 | 20.91 | 317.67 | <0.0001 |
| Residual | 0.92 | 14 | 0.066 | | |
| Lack of Fit | 0.67 | 10 | 0.067 | 1.05 | 0.5247 |
| Pure Error | 0.25 | 4 | 0.063 | | |
| Cor Total | 163.21 | 28 | | | |
| R-Squared | 0.9944 | | | | |

**Table 5.** *Cont.*

| Source | Sum of Squares | df | Mean Square | F Value | p-Value |
|---|---|---|---|---|---|
| Adj R-Squared | 0.9887 | | | | |
| Pred R-Squared | 0.9740 | | | | |
| Adeq Precision | 44.035 | | | | |

The "Adeq Precision" of 44.035 indicated an adequate signal, this model could be used to navigate the design space [23]. The F-value (1.05) and *p*-value (0.5247) of "lack-of-fit" implied that the "lack-of-fit" was not significant relative to the pure error. A not significant lack of fit suggests that the model fits the test data well and has a high reliability. The "Adj R-Squared" of 0.9887 is in reasonable agreement with the "Pred R-Squared" of 0.9740. The R-Squared value of 0.9944 was higher than 0.80, indicating that the accuracy and general availability of the polynomial model were adequate. The very low value of the coefficient of variation (CV%, 2.76) suggested that the experimental values were highly precise and reliable.

In this study, A, C, D, and CD are significant model terms. According to the results, the response surface model constructed in this study for the optimization of cellulase production was considered reasonable.

The final equation of the regression model in terms of coded factors is below:

$$\text{Cellulase activity (U/mL)} = 13.20 + 0.18 \times A - 0.038 \times B + 1.77 \times C + 0.75 \times D - 0.22 \times AB - 0.072 \times AC + 0.12 \times AD + 0.024 \times BC + 0.0053 \times BD - 0.37 \times CD - 1.98 \times A^2 - 1.86 \times B^2 - 4.10 \times C^2 - 1.01\ D^2$$

The final equation of the regression model in terms of actual factors is below:

$$\text{Cellulase activity (U/mL)} = -361.46 + 13.77 \times A + 4.56 \times B + 136.10 \times C + 8.88 \times D - 0.045 \times AB - 0.14 \times AC + 0.15 \times AD + 0.0097 \times BC + 0.0014 \times BD - 0.98 \times CD - 1.98 \times A^2 - 0.074 \times B^2 - 16.40 \times C^2 - 1.70\ D^2$$

### 3.4.2. Optimization of the Cellulase Production

In this study, we utilized Design Expert 10.0.1 for plotting response surfaces to assess the effects of parameters and their interactions on cellulase production.

Figures 11–16 showed six 3D response surface plots and their respective contour plots.

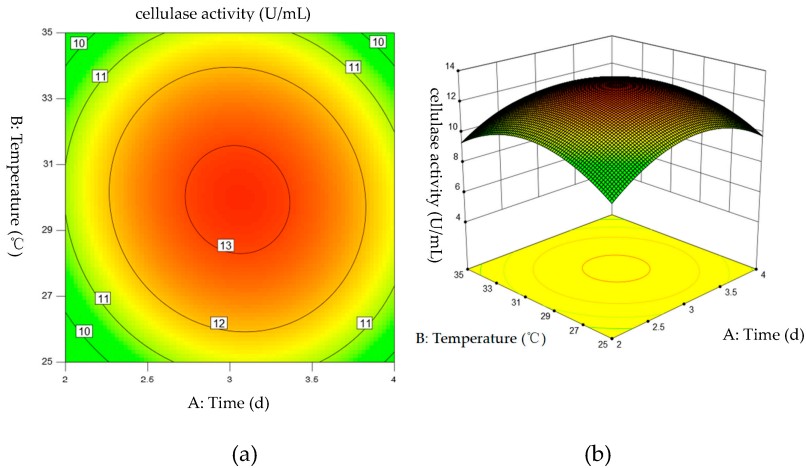

(a)  (b)

**Figure 11.** Contour plots (**a**) and response surface (**b**) representing the effect of time (A), temperature (B), and their reciprocal interaction on cellulase production.

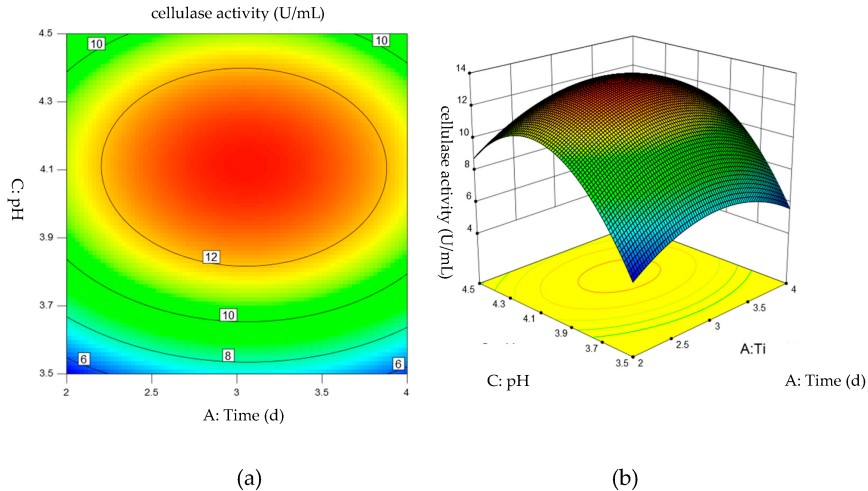

(a)                (b)

**Figure 12.** Contour plots (**a**) and response surface (**b**) representing the effect of time (A), pH (C) and their reciprocal interaction on cellulase production.

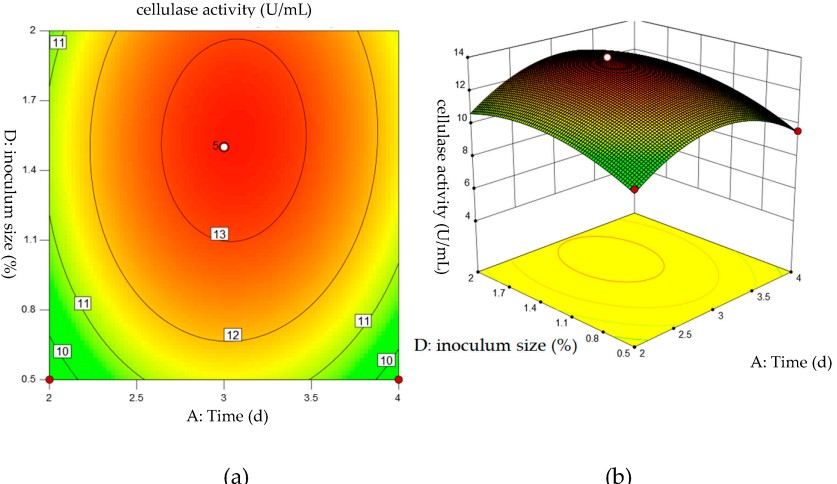

(a)                (b)

**Figure 13.** Contour plots (**a**) and response surface (**b**) representing the effect of time (A), inoculum size (D) and their reciprocal interaction on cellulase production.

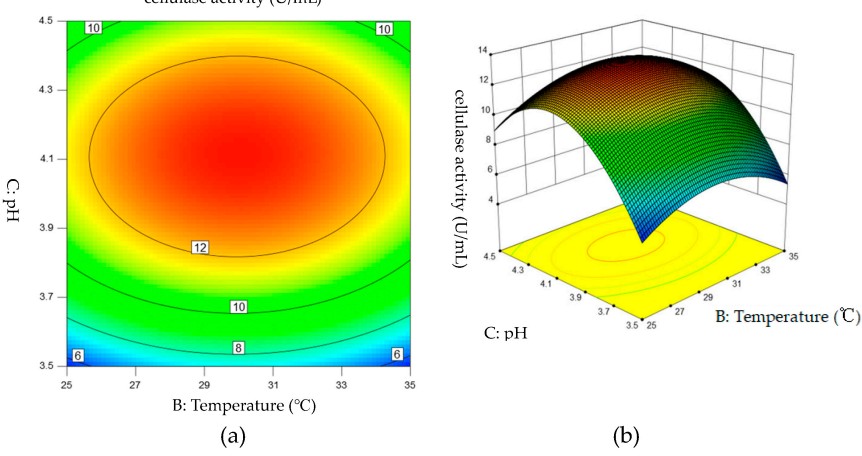

(a)                (b)

**Figure 14.** Contour plots (**a**) and response surface (**b**) representing the effect of temperature (B), pH (C) and their reciprocal interaction on cellulase production.

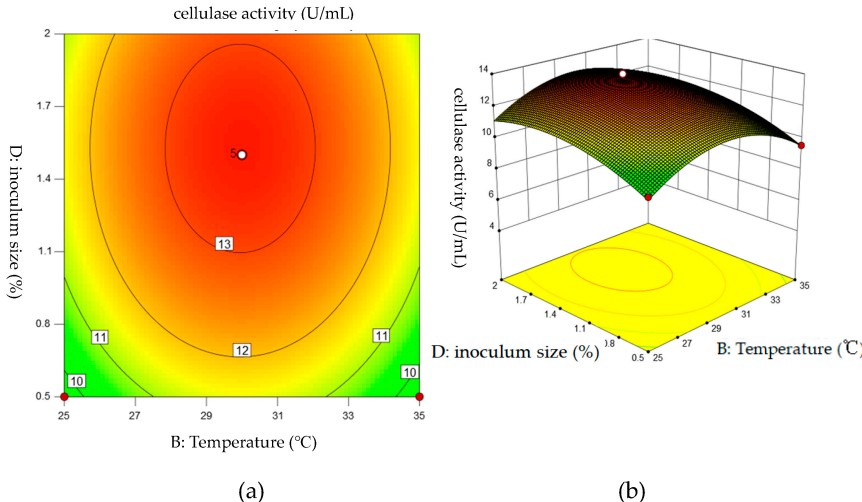

**Figure 15.** Contour plots (**a**) and response surface (**b**) representing the effect of temperature (B), inoculum size (D) and their reciprocal interaction on cellulase production.

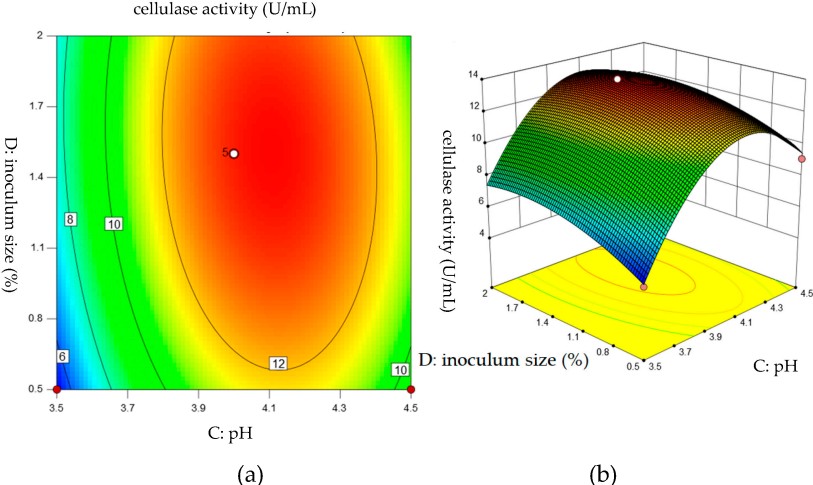

**Figure 16.** Contour plots (**a**) and response surface (**b**) representing the effect of pH (C), inoculum size (D) and their reciprocal interaction on cellulase production.

The 3D response surfaces show interactions of two variables, while the other two were kept constant at their respective zero level.

Figure 11 showed the effect of incubation time (A), incubation temperature (B) and their reciprocal interaction on cellulase production, when pH value (C) was kept at 4 and Inoculum size (D) was kept at 1.5%. The results implied that the cellulase activity increased at first and then decreased with the increasing incubation time (A) and incubation temperature (B).

Similarly, Figures 12–16 showed that the cellulase activity increased at first and then decreased with the increasing of two respective variables.

### 3.4.3. Confirmatory Tests

The optimum cellulase production conditions (A = 3.1 d, B = 29.9 °C, C = 4.1, and D = 1.5%) were estimated using through solving the regression equation and analyzing the response surface contour plots. Predicted cellulase activity was 13.503 U/mL according to the fitted equations under optimized operational conditions.

Three additional experiments were subsequently performed to confirm the prediction. A mean value of 13.44 U/mL of cellulase activity was obtained from the independent laboratory experiments, which agrees well with the predicted response value.

*3.5. Morphological Observation of Substrate Rice Straws before and after Incubation under Different pH Conditions*

After a three-day treatment by strain B13-2 at pH 4.1 and pH 7, the SEM images of rice straws are shown in Figure 17(a,b.1,c.1) showed that the surface of rice straws was flat and smooth with a tight structure and covered by a waxy layer when it was not treated by the B13-2. After a three-day treatment of the B13-2 at neutral conditions, the straws began to show no obvious signs of erosion (Figure 17(b.2)). After a three-day treatment of the B13-2 at a pH of 4.1, the surface of the rice straws was uneven and showed larger holes than in the neutral condition, and the surface was damaged to a greater extent (Figure 17(c.2)). The comparison of the results indicated that the B13-2 has a destructive effect on rice straws and is able to degrade its cellulose tissue, and it is more effective under the pH 4.1 condition.

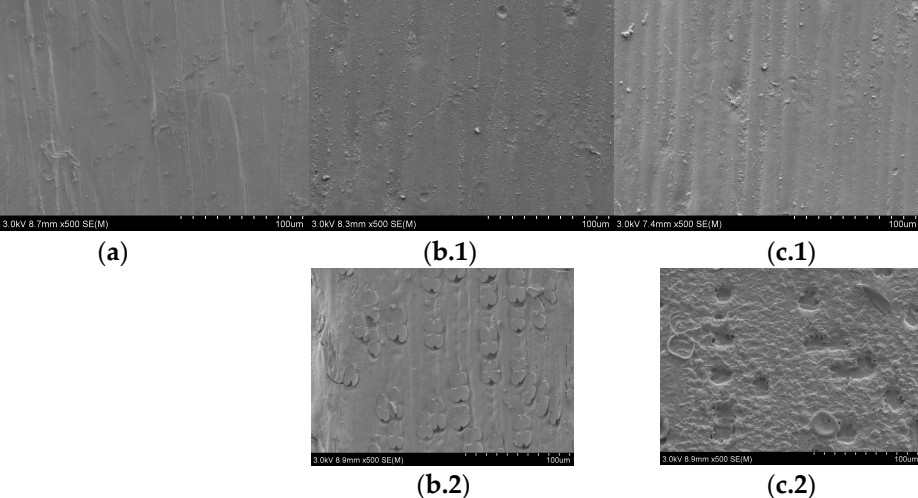

**Figure 17.** SEM images of rice straws under different treatments on day 3. (**a**): the straws without any treatment, (**b.1**): the straws under pH 7.0 without the B13-2, (**b.2**): the straws treated by strain B13-2 at pH 7.0, (**c.1**): the straws under pH 4.1 without the B13-2, (**c.2**): the straws treated by strain B13-2 at pH 4.1.

**4. Discussion**

Cellulase is an important industrial enzyme and plays a major role in many industries. It is able to biologically convert cellulosic biomass into fermentable sugars and the fermentable sugars can be further exploited in many applications [24]. Nowadays, many kinds of cellulase have been screened from bacteria. Cellulolytic bacteria are widespread in the natural environment. We isolated five strains of cellulolytic bacteria from the collected samples and selected one strain (labeled as B13-2) with the highest cellulase activity (in an acidic environment at pH 5.0) as the research object of subsequent further experiments. We identified the cellulolytic strain B13-2 belonging to genera *Raoultella terrigena*.

The fermentation of cellulase production is a very complex biochemical reaction process, and its process is influenced by many factors, such as temperature, incubation time, pH, inoculation rate, carbon, nitrogen sources, etc. This process should be determined by an in-depth study in order to examine the factors affecting the synthesis of the products and to optimize the fermentation process conditions.

Incubation time plays a critical role in enzyme production. With the extension of incubation time, the enzyme system secreted by microorganisms accumulates in the fermentation broth, and the enzyme activity usually increases with time over a certain period, but a too-long incubation time will lead to a decrease in enzyme activity [25]. We gained optimal cellulase production after three days of incubation. After three days, the cellulase activity decreased rapidly. Furthermore, the results indicated poor stability of the enzyme under that condition, which is similar to the result of Jorgensen et al. [26]. Therefore, to find the optimal enzyme production time is necessary to collect a larger amount of

enzyme solution, especially in the efficient industrial production of cellulase, reduce energy consumption, and increase production value.

Temperature is one of the central factors affecting the growth of bacteria and their metabolism, and it is also a crucial parameter for the enzyme production of microorganisms, which influences the secretion of extracellular enzymes by altering the physical properties of the cell membrane [27]. At lower than optimal temperatures, cell membrane permeability is lower and the amount of enzymes obtained is lower; at higher than optimal temperatures, exceedingly high temperatures inhibit the ability of certain active components in microorganisms, while denaturing or producing conformational changes in the enzymes, making them less capable of producing enzymes [28]. In addition, strain B13-2 produced considerable cellulase when the temperature was in the range of 20~45 °C, indicating that the strain has a relatively wide range of growth temperatures and can adapt to different environmental temperatures.

pH is also one of the key factors affecting the enzyme production capacity, which affects the uptake of nutrients by microorganisms in the way of influencing the nature of the surface charge of the microbial cell membrane and the permeability of the membrane [29]. The optimal pH conditions can stimulate strains' growth, increase cellulase production, and maintain the negative feedback mechanism of the enzyme [30]. Strain B13-2 had the highest cellulase production at optimal pH of 4, suggesting that the strain was more inclined to secrete cellulase in an acidic environment. It is very different from other reported strains. Many reported cellulolytic bacteria are fitted for neutral pH conditions, which are from 6.0 to 8.0. However, in the field of agricultural waste processing, it is better that the bacterial or enzymic preparations can withstand acidic environments under certain conditions. The acidophilic and cellulose-degrading properties of strain B13-2 give it potential for application in the agricultural waste treatment process.

Carbon source is the main inducing factor for cellulase synthesis. The selected carbon sources are natural cellulose materials and agricultural wastes with a high yield and low utilization rate, which cause the problem of environmental pollution. Using natural cellulose as the carbon source for cellulase production not only solves the environmental problems, but also greatly reduces the cost of cellulase production, bringing obvious social and economic benefits. In this study, when rice straws were used as the carbon source, strain B13-2 showed the highest cellulose-degrading capacity. It is indicated that the rice straws are the optimal substrate. The cellulase activity and the degradation efficiency are in the highest level which is in the optimal substrate. Therefore, it is again proven that B13-2 can be a new potential engineering bacterial strain to utilize in the agricultural waste treatment process.

Nitrogen is the main factor in the composition of proteins and nucleic acids. The nature of enzymes is protein, and the growth and metabolism of microorganisms cannot be separated from nitrogen sources. Although the nitrogen source does not act as an inducer of cellulase production, it plays an important role in the rapid growth of the bacterium [31,32]. Peptone could boost the cellulose degradation capacity of the strain B13-2 and similar results have been obtained by other researchers [31,32]. Since the metabolism of organic nitrogen sources contributed to the acidification of the medium, higher cellulase production could be gained with organic nitrogen sources than inorganic nitrogen sources [31,32].

The inoculum size could generally limit the growth, metabolism, and the reproduction rate of the bacteria in the fermentation system. The optimal inoculation size of the system can not only shorten the time to reach the peak density in the fermentation system, but also facilitate the formation of products and the use of fermentation substrates and reduce the chance of contamination by miscellaneous bacteria as well. As shown in the results, inoculation size in the range of 1~2% was better for cellulase production of strain B13-2. High inoculation size will lead to insufficient dissolved oxygen in the fermentation system, which is not conducive to the synthesis of the product and the bacterium is prone to senescence, if the inoculation size is too little, it will prolong the fermentation cycle and affect the yield [33].

To further optimize the fermentation conditions, we applied response surface methodology that could evaluate the interactions of the effect of multiple independent factors on cellulase production based on the preliminary range of the parameters determined by the single-factor test for cellulase production. The optimum conditions (time = 3.1 d, temperature = 29.9 °C and pH = 4.1 and inoculum size = 1.5%) for the cellulase production of strain B13-2 were established by solving the regression equation. According to the fitted equations, 13.503 U/mL of cellulase activity was predicted under optimum conditions, which was about 2.4 times as much as before optimization.

To verify that strain B13-2 has better cellulose degradation capacity under acidic conditions than other conditions, we compared the SEM images of rice straws which degraded in buffer with strain B13-2 at pH 4.1 and pH 7.0 separately after three days incubation. The comparison of the results indicate that this strain B13-2 has a destructive effect on rice straws, is able to degrade its cellulose tissue, and is more effective under the pH 4.1 condition.

## 5. Conclusions

This study aimed to screen different kinds of cellulolytic bacteria which can adapt to extreme environments and optimize their cellulase production. One acidophilic cellulolytic bacterial strain was isolated from soil samples and identified as *Raoultella terrigena* strain B13-2. Response surface quadratic model was reliable in the prediction of cellulase production during the fermentation process with the B13-2 strain. The highest level of cellulase production occurred at 29.9 °C, with a pH of 4.1, inoculum size of 1.5%, and 3.1 days of incubation. The most crucial point is that the optimal natural carbon source for this acidophilic cellulolytic B13-2 strain to produce cellulase is rice straws.

Combining these tests, the acidophilic cellulolytic bacteria B13-2 has the potential to become an engineering bacterial strain for the treatment of agricultural waste straw resources. It can process green straw storage and the degradation of straw cellulose. The biggest problem is that the activity of the cellulase produced by this strain is not high enough. In subsequent research, the continuous improvement of fermentation conditions and significant improvements in the cellulase activity are necessary.

Strain B13-2, which has an efficient cellulose degradation ability, exhibited the strongest cellulase production ability in an acidic environment (pH 4.1) and achieved the highest cellulase activity of 13.44 U/mL. It has the potential to be developed into a silage agent. This study of strain B13-2 also provides a foundation for future research on the catalytic mechanism of cellulase from bacteria and the survival mechanism of cellulolytic bacterial cells in acidic environments.

**Author Contributions:** Conceptualization, J.Z.; data curation, J.Z.; formal analysis, S.Z.; funding acquisition, J.Z.; methodology, S.Z.; project administration, J.Z.; software, S.Z.; supervision, X.C.; validation, Z.W.; visualization, J.S.; writing—original draft, S.Z.; writing—review and editing, X.C. All authors have read and agreed to the published version of the manuscript.

**Funding:** This work was funded by the National Natural Science Foundation of China (No. 31300161).

**Institutional Review Board Statement:** Not applicable.

**Data Availability Statement:** Data are available in a publicly accessible repository.

**Acknowledgments:** We thank Microalgae Bioenergy Laboratory (HBUT) for providing valuable information. All authors participated in the writing of the manuscript and agreed with the final format.

**Conflicts of Interest:** The authors declare no conflict of interest.

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
