# Peer review of "Isolation of an Acidophilic Cellulolytic Bacterial Strain and Its Cellulase Production Characteristics"

_agriculture, doi:10.3390/agriculture13071290_

Round 1

Reviewer 1 Report

The article is interesting but in present form it needs several improvements/revisions/explanations, as below:

line 12-13 - " strain (...) was isolated from soil using the Gram’ iodine method and  cellulase activity assay" - these are not the methods of "isolation" but of the characterization of previously isolated bacteria.

line 46-48 - "Bacteria secrete cellulases with richer components than fungi, which are more likely to exert synergistic effects between enzymes and are more beneficial for cellulose degradation" -the fragment is unclear - probably because of the language style - it should be rephrased.

line 60  - "in neutral enzyme-producing medium" - enzymes are not produced by medium!

lines 83-90 - " in my opinion it is not necessary to give the composition of commercially available, commonly used cultivation media - it would be reasonable only when they were modified. On the other hand - the producers of the media should be given.

line 90 - "the culture broth" - it  should be: "microbial suspension".

line 113-119 - (Bacteria Identification by 16s rDNA Sequencing) - procedures should be described in more detail, with more methodological information. 

several times in methodology:   "bacterial solution" must be replaced with "bacterial suspension".

2.7.3, 2.7.4 and 2.7.5 - in the titles the word "incubation" should be removed - it is out of sense in a present form.

Subsection 2.7 -I suggest that the Authors should consider replacing the description with a flowchart leading to the optimization of the final process conditions to make the single-factor procedure more clear and visible.

On what basis the order of optimization of individual parameters was adopted?

Subsection 2.8 and table 1 - the description is too brief, general and unclear. 

line 204 - "Secondary screening tests were carried out for those four cellulose-degrading bacteria" - what about the fifth aforementioned strain?

Line 212-218 - the data should be given in Methods, not in the Results.

line 414 "engineering bacterial strain" - what the Authors meant by?

The English language should be checked/edited extensively.

Reviewer 2 Report

The article concerns the discovery and description of acidophilic cellulolytic bacterial strain and optimisation of its cellulase production. The experimental setup is good, the article is well written and can be published after some corrections/explanations. 

The major concern is the lack of clearity in the experimental setup. Was the culture media designed to isolate the acidophilic bacteria or was it discovered by chance screening numerous (21 in the text ) strains? If so, please write whether the bacterium was identified first and was known to be acidophilic and the cellulases were also expected to be acidophilic or all cellulase-positive strains were tested for production at different pH? Please explain in the Abstract, Introduction, Materials and Methods and Results. 

Also, the title "Isolation and Screening of an Acidophilic Cellulolytic Bacterial  Strain and Its Cellulase Production Characteristics" contains somewhat redundant words;  I suggest to change it to" Isolation of an Acidophilic Cellulolytic Bacterial  Strain and Its Cellulase Production Characteristics".

Reviewer 3 Report

Authors

1.- In the introduction line 30, add more information about the potential of acidophilic bacteria indicating genera and species

 2.- In point 2.1, line 77, include the geographical location of the sampling site, if possible add photography or satellite image

 3.- In line 114, include conditions of the PCR stages and Primers references, also include the DNA purification method as well as its measurement of DNA quality and quantity

 4.- On line 219, add the ascension number of the isolated and identified strain.

Round 2

Reviewer 1 Report

Thank you for the in-depth revision of the manuscript according to my suggestions.

Reviewer 2 Report

Thanks the authors for their efforts to improve the paper. I think the text is now sufficiently improved and warrants publication. I recommend accepting it in its present form. Congratulations, good job!